# Mention Memory: incorporating textual knowledge into Transformers through entity mention attention

**Michiel de Jong**[*][†]**, Yury Zemlyanskiy**[*][†]**, Nicholas FitzGerald**[‡]**, Fei Sha**[‡]**, William Cohen**[‡]
[†] University of Southern California, [‡] Google

## Abstract

Natural language understanding tasks such as open-domain question answering often require retrieving and assimilating factual information from multiple sources. We propose to address this problem by integrating a semi-parametric representation of a large text corpus into a Transformer model as a source of factual knowledge. Specifically, our method represents knowledge with "mention memory", a table of dense vector representations of every entity mention in a corpus. The proposed model - TOME - is a Transformer that accesses the information through internal memory layers in which each entity mention in the input passage attends to the mention memory. This approach enables synthesis of and reasoning over many disparate sources of information *within* a single Transformer model. In experiments using a memory of 150 million Wikipedia mentions, TOME achieves strong performance on several open-domain knowledge-intensive tasks, including the claim verification benchmarks HoVer and FEVER and several entity-based QA benchmarks. We also show that the model learns to attend to informative mentions without any direct supervision. Finally we demonstrate that the model can generalize to new unseen entities by updating the memory without retraining.

## 1 Introduction

Neural models have greatly advanced the state of the art in natural language processing and generation tasks. Accordingly, there has been increasing interest in applying neural language models to tasks which require extensive world knowledge to solve (Petroni et al., 2021). Much of this world knowledge can be found distributed over text corpora, which raises the question whether language models pre-trained on text corpora capture this information. Recent work suggests that while language models may successfully predict facts about the world (Petroni et al., 2019) such knowledge is superficial and unreliable (Cao et al., 2021). Our goal is to reliably incorporate information from across a text corpus into a language model.

Recent work has represented the information present in a text corpus explicitly by constructing a virtual knowledge base (VKB) (Dhingra et al., 2020; Sun et al., 2021). A VKB consists of dense representations of entity mentions in the text, designed to reflect the property or relation expressed by the entity mention. We propose to incorporate a VKB into a language model by using it as an external memory, performing attention over the entire VKB *within* a Transformer model. In this way the model can synthesise and reason over many disparate sources of information from the text corpus. We refer to the VKB used in such a way as Mention Memory, and the model as TOME (Transformer Over Mention Encodings). We first pre-train a mention encoder to specifically encourage mention representations that are useful for a Transformer model, and construct a Mention Memory from 150 million entity mentions in English Wikipedia. Then we train a TOME model with attention layers over the Mention Memory, which is kept frozen (see Figure 1).

We argue that the Mention Memory approach has several appealing properties. *First*, TOME retrieves entity mention representations corresponding to specific entity attributes or relations described in the

---

[*]Equal contribution. Correspondence to {msdejong,yury.zemlyanskiy}@usc.edu. Work primarily done at Google Research. Code is released at `https://github.com/google-research/language/tree/master/language/mentionmemory`

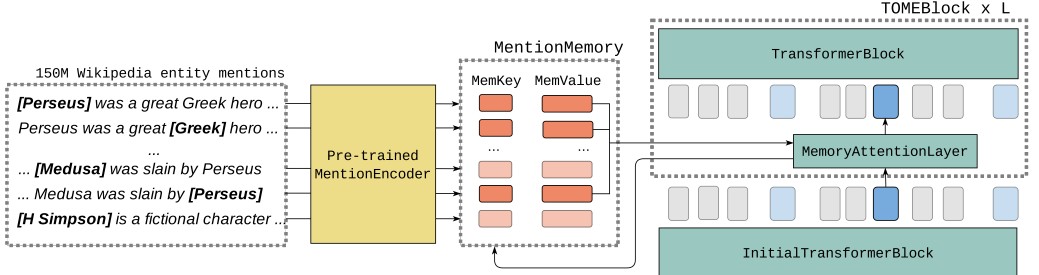

Figure 1: Overview of Mention Memory. A pre-trained mention encoder is used to generate dense representations for each entity mention in Wikipedia (approximately 150 million total) which are stored in a table. The TOME model takes a passage annotated with entity mention boundaries as input, and applies a Transformer block. Next, the TOME model applies one or more TOMEblocks. Each TOMEBlock contains a memory attention layer and a Transformer block.

corpus. This retrieval is much more fine-grained than aggregate entity retrieval methods such as Entities as Experts (EaE) (Févry et al., 2020), and we show large improvements in accuracy over EaE on tasks that require detailed entity information, such as claim verification and entity-based question answering. The fine-grained retrieval also allows potential users to see more precisely what knowledge the model's predictions is based on (see Table 4). *Second*, TOME retrieves dense representations, which are easy to incorporate into a Transformer model without reprocessing the input, unlike raw text. Therefore, TOME is able to retrieve, assimilate and reason over information from many different sources within a *single* Transformer model, allowing for multi-source and multi-hop reasoning without the beam search machinery that is required for multi-hop retrieve-and-read (Zhao et al., 2021). This also makes TOME much more scalable: retrieve-and-read approaches have to read many retrieved passages which becomes expensive with larger reader models, while the cost of memory layers does not scale with reader size and is negligible for larger readers. *Third*, the retrieval is latent, without direct or distant supervision on the retrieved results. We show that, even without supervision, the model learns to retrieve highly specific and informative entity attributes and perform multiple reasoning steps. *Finally*, the memory table is semi-parametric, so knowledge can be added or updated by applying the mention encoder to new text without retraining.

In order to verify the model's capacity to capture accurate factual information in the corpus, we start by evaluating TOME on the HoVer (Jiang et al., 2020), FEVER (Thorne et al., 2018) and FM2 (Eisenschlos et al., 2021) claim verification datasets, on which it strongly improves performance over entity aggregate and comparable retrieve-and-read baselines. We demonstrate that the model learns to attend to informative mentions for verifying claims using only the verification accuracy as a signal. Ablations show the memory is crucial for performance, and that the model can effectively use larger memory than it was pre-trained on. In a second set of experiments we evaluate TOME on question-answering benchmarks TriviaQA (Joshi et al., 2017), ComplexWebQuestions (Talmor & Berant, 2018) and EntityQuestions (Sciavolino et al., 2021), improving performance over comparable baselines. Finally we show that the model can be adapted to generalize to new unseen entities by updating the memory, without retraining.

## 2 METHOD

Our method represents knowledge in a corpus as a collection of "*mention encodings*" – dense vector representations for every entity mention that appears in the corpus. Every time an entity appears in a passage – "*[Barack Obama] was elected president in 2008*" – some property of the entity or its relation to other entities is described. The first component of our method, the Mention Encoder model, is responsible for distilling information from entity mentions in the corpus into high-dimensional mention encodings. We use the Mention Encoder to encode each entity mention in English Wikipedia and gather encodings into a *Mention Memory*. The purpose of the Mention Memory is to capture all knowledge contained in the corpus in a way that can be easily integrated into a Transformer. The second component of our method, the TOME model, applies sparse attention over the Mention Memory to incorporate external information from the corpus into a Transformer model. An overview of the whole method is shown in Figure 1.

Jointly training the Mention Encoder and TOME models is computationally costly, since it would require backpropagating through the Mention Encoder for each attended mention. Consequently, we propose to train the models in two stages. First, we pre-train the Mention Encoder and generate the Mention Memory. Second, we pre-train the TOME model while keeping the Mention Memory frozen: the gradient does not propagate through it and the memories are not modified. Mention Encoder pre-training is specifically designed such that mention encodings capture relevant contextual information about each mention and are useful for TOME even without joint training. We formally define these models in sections 2.1 and 2.2, and their pre-training procedures in 2.3 and 2.4.

**Notation.** An input to the model is a passage $\mathbf{x} = x_1, \ldots, x_T$ of length $T$. We assume that each passage has been annotated with an NER system. Following Baldini Soares et al. (2019) we use special entity markers to highlight entity mentions in the passage. We introduce tokens $[E_{start}]$ and $[E_{end}]$ to the vocabulary and insert them before and after each mention in the passage. For example, the original passage "*What is the nationality of the hero who killed Medusa*" turns into "*What is the $[E_{start}]$ nationality $[E_{end}]$ of the $[E_{start}]$ hero $[E_{end}]$ who killed $[E_{start}]$ Medusa $[E_{end}]$*". Each mention $m$ in a passage is described by a tuple $(s, e)$, where $s$ and $e$ are start and end positions of the mention. We consider entity markers to be part of the corresponding mention, so that $x_s = [E_{start}]$ and $x_e = [E_{end}]$. Representations of these tokens are later used to generate mention encodings.

## 2.1 CONSTRUCTING MENTION MEMORY FROM CORPUS

### 2.1.1 MENTION ENCODER

Let $H \in \mathbb{R}^{T \times d}$ be token representations where $d$ is the hidden dimension, such that $H_i \in \mathbb{R}^d$ is the contextualized embedding for the $i$-th token. Following Févry et al. (2020) we compute the encoding of a span $(s, e)$ as a learnable linear projection W of the concatenation of its start and end token representations $H_s$ and $H_e$

$$\texttt{SpanEncodingLayer}(H, (s, e)) = W[H_s; H_e] \tag{1}$$

The Mention Encoder is a Transformer model with two final `SpanEncodingLayers` that produce *key* and *value* mention encodings. *Value mention encodings* store context-level information about each mention and are used as inputs to the TOME model. *Key mention encodings* identify the type of information stored in the value encodings and serve as attention keys for the memory layer. These two `SpanEncodingLayers` do not share weights.

### 2.1.2 MENTION MEMORY

After the Mention Encoder is pre-trained (see section 2.3), we use it to generate a Mention Memory from entity mentions in Wikipedia. While we could include encodings of any corpus mention in the Mention Memory, we focus on grounded mentions which can be linked to Wikipedia entities. We denote these as *linked* mentions, which we hypothesize contain information that can be retrieved and grounded. We gather mention encodings into matrices `MemKey` $\in \mathbb{R}^{N \times d_K}$ and `MemValue` $\in \mathbb{R}^{N \times d_V}$, where $N$ is the total number of linked entity mentions in English Wikipedia (approximately 150 million) and $d_K$ and $d_V$ are dimensions of key and value encodings. Additionally, we record entity (Wikipedia) IDs of mentions in `MemEnt` $\in \mathbb{R}^N$, which we use as *labels for auxiliary losses*, not as inputs to the model or supervision on retrieval. `MemKey`$(i)$, `MemValue`$(i)$, `MemEnt`$(i)$ correspond to the key encoding, value encoding and entity ID for the $i$-th linked mention in Wikipedia.

## 2.2 TOME MODEL

The TOME model incorporates information from a text corpus into a Transformer by applying sparse attention over the Mention Memory. The model consists of one or more `TOMEBlocks`, each containing a memory attention layer followed by a post-processing Transformer block. Memory attention layers retrieve and attend to relevant "memories" for every mention in the input passage. The model then processes the retrieval-augmented representation with the Transformer block, allowing it to access and combine information from multiple sources in the corpus. Finally, multiple `TOMEBlocks` enable the model to refine retrievals and perform multi-hop reasoning. More formally, a `TOMEBlock` receives the output representation of the previous layer $H$ and produces new representations $H'$

$$M = \texttt{MemoryAttention}(H), \tag{2}$$

$$H' = \texttt{TransformerBlock}(M) \tag{3}$$

The TOME model encodes input passages x with the word embedding layer and initial Transformer block and then applies one or more TOMEBlocks

$$H^0 = \text{InitialTransformerBlock}(\text{TokenEmbedding}(\mathbf{x})), \tag{4}$$

$$H^l = \text{TOMEBlock}_l(H^{l-1}), \; l = 1 \ldots L \tag{5}$$

In this work we consider two configurations of the TOME model: TOME-1 and TOME-2, with one and two TOMEBlocks respectively. Each TOMEBlock of TOME-2 contains half as many Transformer layers as in TOME-1 to hold the total number of Transformer layers fixed between models.

### 2.2.1 MemoryAttention

Each memory attention layer is implemented as a sparse dot-product attention layer that takes the output $H$ of the previous Transformer block, incorporates information from the Mention Memory, and returns a representation $M$ (omitting layer indices). Consider a mention $m$ that starts at position $s$ and ends at position $e$. We start by computing its *query mention encoding* $\text{Query}(m)$ by applying a SpanEncodingLayer

$$\text{Query}(m) = \text{SpanEncodingLayer}(H, (s, e)), \tag{6}$$

Query mention encodings are used to retrieve relevant memories from the Mention Memory table. However, applying standard attention over 150 million mention encodings is infeasible. Instead, we first perform approximate nearest neighbor search to retrieve the top-$K$ mentions with the largest dot product between query $\text{Query}(m)$ and key mention encoding from MemKey. We denote the set of these memories as $\text{TopMem}(\text{Query}(m))$. We compute attention over these memories and incorporate the result into the token contextual representation at position $s$

$$\alpha_i \propto \exp(\text{Query}(m) \cdot \text{MemKey}(i)), \; i \in \text{TopMem}(\text{Query}(m)) \tag{7}$$

$$\text{Value}(m) = \sum_{i \in \text{TopMem}(\text{Query}(m))} \alpha_i \cdot \text{MemValue}(i) \tag{8}$$

$$M_s = \text{LayerNorm}(H_s + W_U \text{Value}(m)) \tag{9}$$

where $W_U$ is a learnable matrix of shape $d \times d_V$.

### 2.2.2 SPARSE LARGE-SCALE RETRIEVAL

Approximate nearest neighbor search (ANNS) can be performed cheaply using one of multiple ANNS libraries, for example ScaNN (Guo et al., 2020). We implemented two on-device search methods to avoid the engineering complexity of real-time communication with an ANNS server, though we have verified this is also viable. The first naively computes a simple dot-product between passage queries and memory keys, and was used in our main experiments as it was easiest to implement. We also implemented and will be releasing a much faster version based on CPU ANNS methods. The memory is sharded over devices, so that the device-memory overhead is negligible.

Holding the number of entries in memory fixed, the compute cost of retrieval from memory does not grow with the size of the reader or the dimensionality of the memory values, so that the relative cost of the memory layer becomes smaller with reader size. In particular, the overhead from the memory used in our pre-training setting is small for BERT-Large and up. More details on ANNS implementation and overhead can be found in Appendix C.

### 2.3 MENTION ENCODER PRE-TRAINING

While backpropagating through a Wikipedia-scale mention memory is challenging, it is possible to train smaller-scale memory architectures end-to-end. We take an approach inspired by MARGE (Lewis et al., 2020a) and READTWICE (Zemlyanskiy et al., 2021) which apply cross-attention over documents within a batch. In particular, we process passages in each batch twice. As a first step, the Mention Encoder model generates mention encodings from each passage and aggregates the mention encodings into a batch-wide memory table. In the second step, we apply a TOME architecture that attends to the batch memory, which we call BATCH-TOME. Note that BATCH-TOME is just used for pre-training the Mention Encoder and not evaluated on any downstream tasks.

Mention Encoder and BATCH-TOME are jointly trained end-to-end so that the Mention Encoder is encouraged to produce mention encodings that contain useful information for BATCH-TOME.

We want to make sure the batch memory contains relevant mentions, so we pre-train the models on batches of passages constructed from related Wikipedia articles with high entity overlap. Appendix A.1 provides more details on Mention Encoder data generation. We use the pre-trained Mention Encoder to construct the Mention Memory table from corpus, and use the BATCH-TOME model as the initialization point for TOME-specific pre-training (described in Section 2.4).

**Masked language model.** Our primary pre-training objective is the standard masked language modeling task, with the loss computed based on the output of the second read (BATCH-TOME). To encourage the model to rely on memory, we increase the task's difficulty relative to standard BERT pre-training by masking entity mention tokens more aggressively.

**Coreference resolution.** We wish to encourage the Mention Encoder to represent the entity attributes expressed by entity mentions, so we also employ an entity-oriented pre-training task to the output of BATCH-TOME for which such attribute information is likely to be especially helpful. Unlike Entities as Experts (Févry et al., 2020), BATCH-TOME does not use entity embeddings, so we cannot use the entity linking task. Instead, we apply a related entity coreference resolution objective, which asks the model to predict whether two linked mentions correspond to the same entity based on the similarity of their encodings. Given that entity surface forms are frequently masked, the model needs to instead use the properties of other mentions in the batch to determine which entity it is most compatible with, incentivizing the Mention Encoder to encode such properties. We compute a coreference mention encoding for every linked mention in the batch by applying a separate `SpanEncodingLayer` on the output of BATCH-TOME. The loss is implemented using cross-entropy over dot-product similarity scores. See Appendix A.2 for details.

## 2.4 TOME PRE-TRAINING

As TOME attends to the full Mention Memory instead of in-batch memory, we do not employ the batching procedure from Mention Encoder pre-training, instead sampling Wikipedia passages randomly. For the same reason, we replace the in-batch entity coreference objective by Mention Memory entity coreference, in which the model has to predict which mentions from the Mention Memory share an entity with the input mention. The goal of this auxiliary objective is to incentivize the model to learn to retrieve informative mention encodings to solve the semantically challenging task. Mention Memory entity coreference also allows us to solve tasks like TriviaQA or ComplexWebQA without a decoder by directly predicting the answer entity.

**Entity prediction.** Analogous to batch coreference resolution loss we compute mention encoding $z_m$ using the output of the TOME model. As in section 2.2, $\texttt{TopMem}(z_m)$ returns the top $K$ memories with the largest dot product between the mention encodings $z_m$ and key mention encodings $\texttt{MemKey}$ from the Mention Memory. The score $\texttt{EntProb}(m, j)$ of entity $j$ equals the sum of attention weights of memories corresponding to this entity.

$$\texttt{EntProb}(m, j) = \frac{\sum_{i \in \texttt{TopMem}(z_m)} \exp(z_m \cdot \texttt{MemKey}(i)) \cdot \mathbb{1}\{\texttt{MemEnt}(i) = j\}}{\sum_{i \in \texttt{TopMem}(z_m)} \exp(z_m \cdot \texttt{MemKey}(i))} \tag{10}$$

The final entity prediction is $\arg\max_j \texttt{EntProb}(m, j)$. Entity prediction loss $\mathcal{L}_{ep}(m)$ for a mention $m$ of entity $\texttt{Ent}(m)$ is $\mathcal{L}_{ep}(m) = -\log \texttt{EntProb}(m, \texttt{Ent}(m))$. Total loss equals the average loss over linked input mentions for which at least one memory of the same entity is retrieved.

**Disallowed same passage retrieval.** For each passage in the pre-training corpus, there exist memories corresponding to mentions in the passage generated from the unmasked version of the same passage. In order to prevent the model from 'cheating' by attending to such memories, we set the attention weight for all memories from the same passage to zero.

## 3 RELATED WORK

Our approach lies at the intersection of three lines of work: i) knowledge-augmented language models, ii) employing a text corpus as a virtual knowledge base, iii) retrieve-and-read methods.

**Knowledge-augmented language models.** Entities as Experts (EaE) (Févry et al., 2020) injects information into a Transformer model model with an intermediate attention layer over trainable entity embeddings, which serve as an aggregate representation of entity information in a text corpus. In contrast, TOME attends to a much larger table of mention encodings, allowing for retrieval of more fine-grained information. Attending to mentions as opposed to entity representations also enables TOME to generalize to unseen entities. FiLM (Verga et al., 2021) extends EaE by adding an attention layer over facts from a KB on the output of the Transformer. The fact attention layer enables more fine-grained queries but still retrieves aggregate entity embeddings as values, which are also not reasoned over by the Transformer. KnowBERT (Peters et al., 2019) is similar to EaE, but with entity embeddings generated from a KB instead of trained end-to-end with a text corpus. MARGE (Lewis et al., 2020a) and READTWICE (Zemlyanskiy et al., 2021) incorporate dense representations from other passages within the same batch into a Transformer through sparse top-k attention. The first pre-training stage of our method for training the Mention Encoder is similar to MARGE and READ-TWICE. However, TOME performs global attention over a full corpus, rather than a single batch. Furthermore, TOME attends to a Mention Memory consisting of pre-computed dense representations. Therefore TOME is not limited to downstream task with batches of relevant documents, and does not need to apply an expensive reader model to an entire batch of documents for each input.

**Text corpus as virtual knowledge base.** DrKIT (Dhingra et al., 2020) performs multi-hop question answering by using a text corpus as a virtual knowledge base. Similar to TOME, the authors apply a mention encoder to convert the corpus into a table of mention encodings. A Transformer model encodes the question into dense queries, which are compared with the mention encodings to *traverse* the VKB. Conversely, TOME *retrieves* mention encodings, and then jointly processes them *inside* the Transformer. In follow-up work to DrKIT, OPQL (Sun et al., 2021) uses a FiLM-like approach to access a memory of relation mentions, which are encoded with a self-supervised relation encoder. However, the relation mention encoding combines a mention-specific relation representation with EaE-like entity encodings, so they are less fine-grained than TOME's encodings. Unlike TOME, OPQL also lacks a sparse large-scale retrieval mechanism, and relies on *ad hoc* heuristics to limit the size of the memory.[1] MOLEMAN (FitzGerald et al., 2021) compares a passage mention encoding with mention encodings from a corpus to perform entity linking, but does not retrieve the mentions.

**Retrieve-and-read methods.** REALM (Guu et al., 2020) learns to retrieve relevant passages from a text corpus in a self-supervised manner. Retrieved passages are concatenated to the input passage which is then re-encoded by a Transformer model to perform a task. The key difference between retrieve-and-read approaches (Guu et al., 2020; Karpukhin et al., 2020; Lewis et al., 2020b; Izacard & Grave, 2021) and TOME is that TOME retrieves dense representations, as opposed to text. That means that TOME only applies a reader model once to a single input, while retrieve-and-read approaches have to apply an expensive BERT reader to many different passages. In addition, Transformer models can only process relatively short sequences, which imposes a binding constraint on the number of retrieved text passages that can be processed *together*, whereas TOME can retrieve and reason over information from many sources inside the same reader. Generative models like RAG (Lewis et al., 2020b) or FiD (Izacard & Grave, 2021) attend to different retrieved documents in the decoder, but still have to apply a BERT read for every retrieved document, do not consider interaction between retrievals while encoding the question, and cannot perform iterative retrieval.

## 4 EXPERIMENTS

### 4.1 EXPERIMENTAL SETUP

The Mention Encoder is based on a BERT-base model with two final `SpanEncodingLayers` that produce key and value encodings. Mention Encoder and BATCH-TOME share Transformer weights during Mention Encoder pre-training. The Mention Memory consists of mention encodings for $N = 150$ million linked Wikipedia entity mentions. Transformer layers in TOME and BATCH-TOME models are equivalent to those in the BERT-base model. The TOME `InitialTransformerBlock` contains 4 Transformer layers. TOME-1 has a single `TOMEBlock` with 8 Transformer layers, and TOME-2 has two `TOMEBlocks` with 4 Transformer layers each. Therefore, the number of trainable parameters in TOME-1 and TOME-2 is approximately the same as in BERT-base. We use a smaller

---

[1]It should be noted that absent heuristics, the number of potential relation mentions (i.e., entity mention pairs) is much larger than the number of entity mentions.

Table 1: Accuracy on claim verification datasets. #Encoded refers to the number of passages encoded by a BERT reader to answer a single question.

| Model | #Params | #Encoded | HoVer$_{test}$ | FEVER$_{test}$ | FM2$_{dev}$ |
|---|---|---|---|---|---|
| RAG | 620M | 100 | - | 72.5 | - |
| REALM | 330M | 5 | 66.1 | 67.1 | 65.8 |
| Entities as Experts | 360M | 1 | 66.6 | 63.6 | 63.5 |
| TOME-1 | 220M | 1 | 72.8 | 67.8 | 67.7 |
| TOME-2 | 220M | 1 | 73.1 | 68.1 | 68.4 |

Mention Memory containing 38m uniformly sampled memories for TOME pre-training. During fine-tuning and evaluation we utilize the full Mention Memory. Appendix A contains more details.

## 4.2 BASELINES

We compare TOME with existing methods that utilize textual information from a corpus in a language model. These can be divided into generative LLMs (T5), entity embedding retrieval (Entities as Experts, OPQL), extractive retrieve-and-read (REALM) and generative retrieve-and-read (RAG, Fusion-in-Decoder). TOME occupies a novel position in the space of retrieval models, being more fine-grained than entity embedding retrieval methods, but performing all its reasoning with a single BERT read, unlike retrieve-and-read methods. The most closely comparable models are Entities as Experts and REALM, and we use these as our primary baselines. We report other baselines for reference, with the caveat that these results are not apples-to-apples: RAG and Fusion-in-Decoder have large decoders and retrievers and encode a large number of passages with a BERT reader for each question compared to TOME's single read. Fusion-in-Decoder and RAG[2] also use ground-truth supervision for retrieval. We mark the number of parameters and BERT applications for each baseline in the result tables. Consistent with retrieve-and-read, we count the parameters of the Mention Encoder and TOME, but not the size of the non-trainable and sparsely accessed Mention Memory.

## 4.3 CLAIM VERIFICATION

**Data.** Our first set of experiments evaluates TOME on the claim verification tasks FEVER (Thorne et al., 2018), HoVer (Jiang et al., 2020), and FM2 (Eisenschlos et al., 2021) in which the model is provided with a claim and has to determine whether the claim is supported by the Wikipedia corpus. FEVER is a larger dataset with 186k claims for which most of the claims can be verified with a single Wikipedia passage. In contrast, HoVer is smaller with 26k claims, but is explicitly constructed to require evidence from multiple sources and multiple reasoning steps. FM2 is also smaller and is constructed through an adversarial game that leads to more challenging retrieval. The claim verification training data contains gold evidence passages, but unlike most published results *we do not use these*, leaving only the accuracy of the claim verification to guide the retrieval.

**Results.** Table 1 contains our claim verification results. TOME outperforms both Entities as Experts and REALM, especially on HoVer and FM2. This is consistent with the properties of TOME: HoVer requires combining detailed information from multiple sources, which TOME is especially well equipped to do compared to aggregate entity-based or retrieve-and-read models. FM2 features generally challenging retrieval and may benefit from contextualizing retrieved evidence.

## 4.4 QUESTION ANSWERING

**Data.** In a second set of experiments we evaluate TOME on TriviaQA (TQA) (Joshi et al., 2017), ComplexWebQuestions (CWQ) (Talmor & Berant, 2018) and EntityQuestions (EQ) (Sciavolino et al., 2021), open-domain QA tasks for which most answers are Wikipedia entities. We approach these datasets as entity-linking tasks, as in Févry et al. (2020). We append a mask token to each question, which is marked as a question mention. The probability for each candidate entity is predicted as the aggregate attention weight on mentions of the entity (Section 2.4). Questions with answers that do not correspond to entities in our entity vocabulary are marked as answered incorrectly. TQA

---

[2]RAG is initialized from DPR which is trained with gold retrieval passages for TriviaQA.

Table 2: Accuracy on open-domain QA datasets TriviaQA (TQA), ComplexWebQuestions (CWQ) and EntityQuestion (EQ). #Encoded refers to the number of passages encoded by a BERT reader to answer a question. $TQA_{e\text{-}dev}$ corresponds to TQA with train and dev samples limited to those with Wikipedia entity as an answer. See Appendix B.3 for full results.

| Model | #Params | #Encoded | $TQA_{dev}$ | $TQA_{test}$ | $TQA_{e\text{-}dev}$ | $CWQ_{dev}$ | $EQ_{dev}$ |
|---|---|---|---|---|---|---|---|
| RAG | 620M | 100 | 56.8 | 68.0 | - | - | - |
| Fusion-in-Decoder | 440M | 100 | 65.0 | 77.1 | - | - | |
| REALM | 330M | 5 | 55.8 | 67.1 | 63.4 | 46.7 | 59.0 |
| T5-3B | 3B | 1 | - | - | - | 38.7 | - |
| T5-11B | 11B | 1 | 42.3 | 50.1 | - | - | - |
| Entities as Experts | 360M | 1 | 43.2 | 53.4 | 51.3 | 42.7 | 32.5 |
| OPQL | 220M | 1 | - | - | - | 41.1 | - |
| TOME-1 | 220M | 1 | 50.8 | 61.1 | 60.3 | 44.9 | 62.1 |
| TOME-2 | 220M | 1 | 54.6 | 65.8 | 64.8 | 47.7 | 66.0 |

Table 3: TOME-2 retrievals for the second HoVer dev sample. We show top-1 retrieval results for the first ($\longrightarrow_1$) memory attention layer for two passage mentions. Memory mentions are in brackets.

| |
|---|
| Claim: **Greater Swiss Mountain Dog** and **Harrier** are both **dog breeds**. Label: TRUE |
| **Greater Swiss Mountain Dog** $\longrightarrow_1$ Breed History the origin of the **[Greater Swiss Mountain Dog]** is not definitively known. . . . |
| **Harrier** $\longrightarrow_1$ The harrier is a medium-sized dog breed of the **[hound]** class, used for hunting. . . |

consists of 96k trivia questions, for which 84% of answers correspond to a Wikipedia entity. We use the open-domain setting without gold evidence passages. In order to compare head-to-head performance, we also report results on a subset of TQA with only questions with Wikipedia entities as an answer. CWQ consists of 35k complex questions (compositions, conjunctions, etc.) for which 94% of answers correspond to a Wikipedia entity. EQ contains challenging questions involving rare entities, with Wikipedia entities as answers.

**Results.** Table 2 contains the results for TQA, CWQ and EQ experiments. Like TOME, Entities as Experts and OPQL treat the above datasets as entity-linking tasks. REALM performs extractive QA, while T5, RAG and Fusion-in-Decoder generate the answer. We note a similar pattern of results as for claim verification. TOME strongly outperforms Entities as Experts on all tasks. TOME performs slightly better than REALM on a simple task like TriviaQA (entity subset) and strongly outperforms REALM on more challenging tasks that require multiple (CWQ) or challenging (EQ) retrieval.

### 4.5 QUALITATIVE PROPERTIES OF TOME

**What memories does TOME retrieve?** Given that TOME retrieval is unsupervised, it is natural to ask what memories it learns to retrieve. First, we observe that BATCH-TOME and TOME trained on just the MLM objective learn to attend to memories of the same entity as the passage linked mention (55% and 41% average attention score). This is promising as entity mentions from the same entity often contain mutually relevant information. Quantitative evaluation of downstream retrieval is challenging as TOME often retrieves mentions that are not part of, but equally informative as gold passages. Instead, we provide TOME retrievals on *the first three* samples of the HoVer dev set to demonstrate its retrieval behavior without cherry-picking. Table 3 demonstrates a successful simple retrieval, while Table 4 displays interesting multi-hop retrieval. The last is found in Appendix D.

**Importance of memory size.** Figure 2 shows claim verification performance as a function of memory-size during fine-tuning (pre-training memory size is held constant). For smaller memory sizes, entries in memory are uniformly sampled from the full Mention Memory. Performance increases smoothly with memory size. Larger memory size yields diminishing returns, perhaps reflecting that entity mentions may contain overlapping information.

Table 4: TOME-2 retrievals for the first HoVer dev sample. We show top-1 retrieval results for the first ($\longrightarrow_1$) and the second ($\longrightarrow_2$) memory attention layers for passage mentions "Life Goes On" and "Hungry"[3]. Memory mentions are in brackets. The first retrieval for the "Life Goes On" is a different song with the same name and the first retrieval for "Hungry" is related but not useful. However, the second retrieval for "Life Goes On" identifies the correct song and describes its position on the album while the second retrieval for "Hungry" captures its position relative to "Life Goes On".

---

Claim: **The song** recorded by **Fergie** that was produced by **Polow Da Don** and was followed by **Life Goes On** was **Hungry**. Label: TRUE

---

**Life Goes On** $\longrightarrow_1$ . . . and Johnny J produced the chart topping hits "All Bout U", "How Do U Want It" and **["Life Goes On"]**. . . .
**Life Goes On** $\longrightarrow_2$ . . . On November 11, 2016, Fergie released the third single from the album, **["Life Goes On"]**. . .

---

**Hungry** $\longrightarrow_1$ . . . Polow da Don, is an American record producer, songwriter and rapper. His cousin is **[Atlanta]** singer Monica. Jones has produced a variety of singles for a multitude of artists including "Anaconda" by Nicki Minaj (2014), "Love In This Club" by Usher (2008), "Buttons" by the Pussycat Dolls (2006), "Hungry" by Fergie . . .
**Hungry** $\longrightarrow_2$ . . . "Life Goes On" is a song recorded by American singer Fergie for her second studio album, Double Dutchess (2017). . . . The song serves as the third single from **[Fergie's]** second studio album, following "Hungry".

---

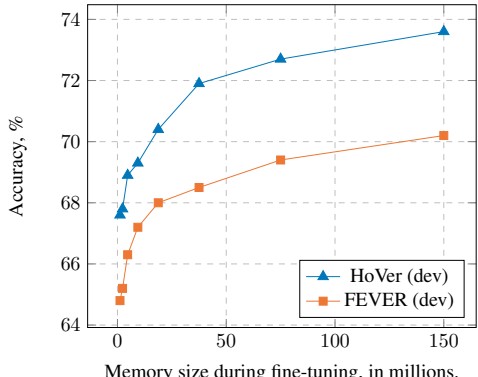

Figure 2: Claim verification accuracy as a function of fine-tuning memory size (in millions).

Table 5: Accuracy on held-out subset of TriviaQA and ComplexWebQuestions (CWQ) questions. TOME-1-unseen was pre-trained and fine-tuned with memory without entities from held-out set and evaluated with full memory. Note that performance is considerably lower than on the full dev set as answers in the held-out set (which are in dev but not train) are more likely to be rare entities.

| Dataset | TriviaQA$_{dev}$ | CWQ$_{dev}$ |
|---|---|---|
| TOME-1 | 17.4 | 16.4 |
| TOME-1-unseen | 17.6 | 16.7 |

**Zero-shot transfer to unseen entities.** An important advantage of memory architectures is that the behavior of the model can be steered by deciding what to include in the memory. Here we show that the TOME model can use information that was not present in memory during training. We sample questions in the TQA and CQA dev sets, and generate a subset of the memory without any mentions corresponding to the answer entities for those questions. Then we pre-train and fine-tune a model on this smaller memory, which we call TOME-unseen. We evaluate TOME-unseen on the sampled questions *using the full memory for evaluation only*, and compare to standard TOME. Table 5 shows that using full memory only during evaluation does not lower performance.

## 5 CONCLUSION

We introduced TOME, a Transformer model that performs attention over a semi-parametric representation of the entire Wikipedia text corpus. This representation, or Mention Memory, consists of a dense encoding for each entity mention in Wikipedia. TOME can retrieve information from multiple sources without supervision, aggregate information within the Transformer, and reason over the retrieved information. TOME leads to strong improvements on multiple open-domain claim verification and entity-based question answering tasks.

---

[3]We replaced the original song title with the song "Hungry" as the original may be inappropriate.

**Ethics.** We pre-train and evaluate on public data without privacy-sensitive information. We aim to improve performance on established academic NLU tasks, which may eventually extend to performance on tasks with more direct ethical implications. Improving the ability of Transformers to incorporate textual knowledge may help with reducing factual errors and hallucinations for large language models, which are known to carry risks.

**Reproducibility.** We have release coded for the model and pre-training, as well as pre-trained TOME-1 and TOME-2 model checkpoints and corresponding Mention Memory. Appendix A provides details of the pre-training procedure. The most challenging part of the model to reproduce is the sparse attention layer, described in Appendix C. We do not release pre-training data, but the data can be easily reproduced from Wikipedia dumps and the publicly available Google Cloud API.

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

## A  PRE-TRAINING

We train on English Wikipedia, processed with the entity linking and named entity recognition tools from the Google Cloud NLP API[4]. We use existing hyperlinks in Wikipedia as additional entity annotations. All models are pre-trained on 128 TPUs using AdamW optimizer (Loshchilov & Hutter, 2019) with learning rate 1e-4 and batch size of 4096. Each passage in the batch has length $T = 128$, excluding entity tokens. The Mention Encoder and BATCH-TOME are pre-trained for 1 million steps with 50k warmup steps, and TOME is trained for 500k additional steps with 25k warmup steps after initialization from BATCH-TOME. Both models are trained with linear learning rate decay. Mention Encoder and BATCH-TOME share Transformer weights during Mention Encoder pre-training. We apply gradient clipping with a norm of 1.0 and weight decay of 0.01. Weight decay is applied to all weights except layer norm and bias weights.

BATCH-TOME and TOME are trained with weight 0.85 on the MLM objective and 0.15 on the entity coreference resolution objective. We mask 20% of whole entity mentions and 10% of other tokens. We limit the coreference resolution objective to mentions of the 1 million most frequent Wikipedia entities. We use 24 mentions per sample, with a batch size of 32 samples per TPU. We subsample mentions uniformly if the average number of annotated mentions on a TPU exceeds 24. Key mention encoding have dimension $d_K = 128$ and value and coreference mention encodings have dimension $d_V = d_C = 512$.

---

[4]https://cloud.google.com/natural-language/docs/basics#entity_analysis

**Disallowed same passage retrieval for Mention Encoder.** We want the model to use memory as a source of additional information for processing a passage. Therefore, we explicitly set attention weights to 0 for memories generated from the same passage as the current one.

### A.1 MENTION ENCODER DATA GENERATION

We pre-train Mention Encoder to produce mention encodings that are useful for BATCH-TOME. In order to provide BATCH-TOME with an incentive to use the memory, we need to ensure that mentions from different samples within a batch are relevant to each other. We achieve this by batching passages from the same or related Wikipedia articles.

We generate clusters of 256 passages from Wikipedia articles using a greedy method. First, we create a cluster from the longest unused Wikipedia article and add related articles until the cluster consists of 256 passages. In particular, at each step we add the article with the largest Jaccard similarity between its entity set and the entity set of articles in the current cluster.

### A.2 COREFERENCE RESOLUTION LOSS

For every linked mention $m$ in the batch we compute a mention encoding $z_m$ by applying a separate `SpanEncodingLayer` on the output of the BATCH-TOME. First, we compute the loss for every linked mention $m$ in the batch. To this end, we denote linked mentions in *every other passage* in the batch as positive, $\mathcal{P}^+(m)$, if they have the same entity ID as $m$ and negative, $\mathcal{P}^-(m)$, otherwise. The loss per mention is an average of cross-entropy losses per every positive mention $m^+ \in \mathcal{P}^+(m)$

$$\mathcal{L}_{coref}(m) = -\frac{1}{|\mathcal{P}^+(m)|} \log \sum_{m^+ \in \mathcal{P}^+(m)} \frac{\exp(z_m^T z_{m^+})}{\exp(z_m^T z_{m^+}) + \sum_{m^- \in \mathcal{P}^-(m)} \exp(z_m^T z_{m^-})}$$

The total loss is average of losses per linked mentions which have at least one positive mention (set $\mathcal{P}^+(m)$ is not empty).

## B EXPERIMENTS

### B.1 FINE-TUNING SETUP

TOME is fine-tuned on 32 TPUs using the Adam optimizer with a learning rate of 1e-5 and total batch size 32. In contrast to pre-training we set max mentions to 32 per sample for fine-tuning. We use 1000 warmup steps and linear learning rate decay. Gradient clipping and weight decay are the same as during pre-training. We take the highest scoring checkpoint on dev sets and evaluate it on the test set. We use the spaCy noun chunker to detect noun phrases and treat these as claim/question entity mentions.

The model can be fine-tuned with full memory on a server of 8 A100 GPUs or 16 v3/v4 TPUs. A model with half memory (75M mentions) can be fine-tuned on 8 V100s/P100s or 8 TPUs.

### B.2 BASELINES

Following Guu et al. (2020) we used REALM to perform extractive question answering on TriviaQA and ComplexWebQuestions datasets. We also adapted the model to the classification setting in order to apply it to claim verification tasks. Given an input claim $X$ we compute probability of a prediction $Y$ (whether the claim holds true or not) as a marginalization over retrieval $Z$.

$$\Pr(Y|X, Z) = \sum_{z \in Z} \Pr(Y|X, Z = z) \cdot \Pr(Z = z|X)$$

where $\Pr(Y|X, Z = z)$ is the output probability produced by the reader model and $\Pr(Z = z|X)$ is produced by the retrieval model.

### B.3 CLAIM VERIFICATION

See Table 6 for the results on development and test splits of claim verification datasets. Additionally, Table 7 compares our FM2 results to the original dataset baselines.

Table 6: Accuracy on claim verification datasets. #Encoded refers to the number of passages encoded by a BERT reader to answer a single question. EaE stands for Entities as Experts model.

| Model | #Params | #Encoded | HoVer$_{dev}$ | HoVer$_{test}$ | FEVER$_{dev}$ | FEVER$_{test}$ | FM2$_{dev}$ |
|---|---|---|---|---|---|---|---|
| RAG | 620M | 100 | - | - | 74.5 | 72.5 | - |
| REALM | 330M | 5 | 67.3 | 66.1 | 70.4 | 67.1 | 65.8 |
| EaE | 360M | 1 | 66.2 | 66.6 | 66.1 | 63.6 | 63.5 |
| TOME-1 | 220M | 1 | 73.6 | 72.8 | 70.5 | 67.8 | 67.7 |
| TOME-2 | 220M | 1 | 74.1 | 73.1 | 71.1 | 68.1 | 68.4 |

Table 7: Accuracy on FM2 compared with original dataset baselines. Oracle refers to oracle retrieval followed by a BERT-Base reader.

| Model | Accuracy |
|---|---|
| Oracle (Eisenschlos et al., 2021) | 69.3 |
| DPR (Eisenschlos et al., 2021) | 64.2 |
| EaE | 63.5 |
| REALM | 65.8 |
| TOME-1 | 67.7 |
| TOME-2 | 68.4 |

## B.4 Question Answering

We report additional results on the EntityQuestions dataset from Sciavolino et al. (2021). The dataset consists of questions involving rare entities, making it especially challenging for modern retrievals methods such as DPR. Evaluation results for TOME models and baselines are shown in Table 8 and Table 9. Following (Sciavolino et al., 2021) we report recall at 20 as an evaluation metric. Since TOME retrieves mentions rather than passages, a direct comparison is difficult. We evaluate TOME conservatively, treating recall at 20 as successful if one of the 20 highest scoring mentions belongs to the correct entity (in contrast to DPR, for which the correct answer only has to be somewhere in the retrieved 100 word document).

TOME sets the state of the art on this dataset and outperforms DPR by a very large margin. REALM cannot be fairly compared to DPR due to longer retrieved passages (100 vs 288 tokens). Therefore, we perform a separate experiment using accuracy with REALM as a baseline, showing large performance gains over REALM as well.

## B.5 Importance of pre-training objectives

We perform several ablation experiments for the pre-training procedure (see Table 10). First, results show that the entity prediction objective (c.f. Section 2.4) is not essential for TOME pre-training. Performance on claim verification datasets (FEVER and HoVer) is not affected by whether we use entity prediction for pre-training. More surprisingly, removing this objective only slightly decreases the performance on entity question answering datasets (TriviaQA and ComplexWebQuestions). We predict entities for question-answering in the same way as we do for the entity prediction objective during pre-training (c.f. Equation 10), so we expected the entity prediction auxiliary loss to be important.

Table 8: EntityQuestions recall@20

| Model | Recall@20 |
|---|---|
| DPR (Sciavolino et al., 2021) | 65.4 |
| BM25 (Sciavolino et al., 2021) | 71.2 |
| TOME-1 | 83.3 |
| TOME-2 | 83.8 |

Table 9: EntityQuestions top-1 accuracy

| Model | Accuracy |
|---|---|
| Entities as Experts | 32.5 |
| REALM | 59.0 |
| TOME-1 | 62.1 |
| TOME-2 | 66.0 |

Table 10: Performance ablations for pre-training objectives experiments.

| Dataset | HoVer$_{dev}$ | FEVER$_{dev}$ | TriviaQA$_{dev}$ | CWQ$_{dev}$ |
|---|---|---|---|---|
| TOME-1 | 73.6 | 70.5 | 50.8 | 44.9 |
| w/o entity coreference loss | 69.8 | 68.4 | 42.5 | 40.5 |
| w/o entity prediction loss | 73.7 | 70.7 | 49.4 | 43.8 |

On the other hand, a related entity coreference objective (c.f. Section A.1 and Appendix A.2) is crucial for Batch-TOME and Mention Encoder pre-training. That is consistent with our intuition that semantically challenging tasks incentivize the model to store useful information in memory.

### B.6 TOME INITIALIZATION

We initialize TOME model with a pre-trained BATCH-TOME model which we find to be especially important for warming up retrieval. If TOME is initialized from scratch (or even from BERT weights), TOME does not learn to use the memory. In fact, TOME has to be initialized from the same BATCH-TOME used to generate the memory. This implies that multi-stage training is a vital ingredient for TOME to succeed. Our explanation for why TOME is sensitive to initialization is that TOME needs to learn two skills: first, to effectively use retrieved mentions for its predictions, and second, to retrieve relevant mentions. Learning both capabilities end to end gives rise to a mutual dependence: to get a signal for learning how to use retrieved mentions, the retrieved mentions have to be useful, and to learn to retrieve useful mentions, the model needs to utilize retrieved mentions. If initialized from scratch, the model is not able to learn both skills simultaneously. The pre-training stage with the smaller in-batch memory functions as a curriculum to address that problem.

## C NEAREST NEIGHBOR SEARCH

Nearest neighbor search is an extremely common problem, and there exist numerous approaches and packages for fast approximate nearest neighbor search (ANNS). (Guo et al., 2020; Johnson et al., 2017). Most approaches employ two methods for fast search: 1) compress the search table through projecting to a lower dimension and quantization and perform comparisons in this compressed space and 2) divide the search table in buckets of similar items, and search only a subset of the buckets. Retrieve-and-read use ANNS packages to search for related passages (Guu et al., 2020; Lewis et al., 2020b).

Applying such packages for TOME is slightly trickier, as TOME needs to perform ANNS inside the model. One viable route is to compute queries on-device, transmit them to a separate ANNS server and them transmit them back. We would recommend this approach for GPU accelerators, with faster host-device communication and slower device-device communication. As we are using TPU accelerators, we decided to use on-device ANNS, which does not require coordinating additional servers and will potentially allow for backpropagating through memory in future work.

### C.1 ON-DEVICE NEAREST NEIGHBOR SEARCH

We shard the Mention Memory over all TPU devices. We perform search by distributing each query to all devices and retrieving top-K search results from each local memory shard. Then, the results are distributed back to the original devices and the local search results are aggregated through another, global top-K.

#### C.1.1 DOT-PRODUCT

The first method we describe is naive dot-product search, taking advantage of matrix multiplication capacity of TPU accelerators. In this method we perform search over local shards by taking the dot product between the query and the local memory shard and performing an approximate top-k operation over the results. Dot-product search is easy to implement and fast for smaller memory sizes (up to 10 million entries). We implemented this method first due to its simplicity and our primary experimental results employ this search method.

### C.1.2 ANNS

To speed up search we implemented method 2) from standard CPU-based ANNS, bucketing the search table and searching only a subset of buckets. In particular we perform k-means clustering to divide the Mention Memory into clusters, and perform dot-product search over the top $n_s$ clusters on each device.

### C.1.3 OVERHEAD

While the Mention Memory is stored on-device, memory overhead is negligible as the memory table is sharded. For pre-training the Mention Memory took up 2.2% of available device memory. Table 11 shows percentage of time spent on ANNS in TOME-1 pretraining for different reader architectures. The relative overhead of search becomes smaller with reader size, and ANNS overhead in particular becomes negligible for BERT-Large and up. We did not measure standard CPU ANNS overhead, but it should be comparable to or faster than our ANNS numbers.

Table 11: Proportion of time spent on ANNS for TOME-1 pre-training setting.

| Model | Dot-product | ANNS |
|---|---|---|
| BERT-Base | 0.79 | 0.22 |
| BERT-Large | 0.48 | 0.07 |
| T5-11B Encoder | 0.17 | 0.02 |

### C.1.4 HYPERPARAMETERS

For ANNS in `TOMEBlocks` we take top-2 search results from each local memory shard, and apply top-128 over the retrieved results. For ANNS in the entity prediction layer we take top-32 search results from each local shard, and aggregate across shards without applying an additional top-K operation.

## D  RETRIEVAL EXAMPLES

Table 12: TOME-2 retrievals for the second HoVer dev sample. We show top-1 retrieval results for the first ($\longrightarrow_1$) memory attention layer for passage mentions "the novel", "the movie" and "the album". Memory mentions are in brackets. We can see that the model can retrieve relevant mentions for non-named passage mentions, and generally understands it is looking for mentions related to music. However, while the best retrieval for "album" is from a passage that mentions sampling the Shining, it is quite far removed and it is likely the retrieval is not sufficiently accurate here.

---

Claim: **Stephen King** wrote **the novel** that **the movie** directed by **Stanley Kubrick** that was sampled in **the album** **"Where Blood and Fire Bring Rest"** was based on. Label: TRUE

---

**the novel** $\longrightarrow_1$ Music Video. The video is a homage to Stanley Kubrick's 1980 film The Shining based on the **[Stephen King]** novel . . .

**the movie** $\longrightarrow_1$ Music Video. The video is a homage to Stanley Kubrick's 1980 film The Shining based on the **[Stephen King]** novel . . .

**the album** $\longrightarrow_1$ Where Blood and Fire Bring Rest is the third full-length album released by **[metalcore]** band ZAO. It was the first album to feature vocalist Dan Weyandt after the departure of Shawn Jonas along with new bassists/guitarists, Russ Cogdell and Brett Detar. The album contains a sample from the film The Shining . . .

---

