# OpenReview forum: "Mention Memory: incorporating textual knowledge into Transformers through entity mention attention"
_ICLR.cc/2022/Conference — ICLR 2022 Poster_

### Official Review · Reviewer_giXh · 2021-10-28

**Correctness:** 4
**Technical Novelty And Significance:** 3
**Empirical Novelty And Significance:** 3
**Recommendation:** 6
**Confidence:** 4

**Main Review:**

Pros:
- a batch-wise mechanism is proposed to support efficient pre-training.
- entity-aware pre-training tasks are proposed to learn embeddings that can better embed entity knowledge.
- evaluations conducted are comprehensive and sold.

Cons:
- seems the proposed model still needs an NER to detect entity mentions from each input passage, which could lead to an unexcepted performance decrease when the model is used for out-domain inputs.
- I suggest to include more entity-insensitive NLP tasks in the paper, to verify that proposed entity-aware mechanism is not biased on entity-related tasks.

**Summary Of The Paper:**

This paper proposes an external knowledge-based pre-trained model that can leverage entity-wise knowledge embeddings. The mention encoder model encodes entity mentions occurred in an external corpus (i.e. Wikipedia) into embeddings as the memory. The TOME model retrieves most possible entity mentions from this memory and performs attention on them to generate aggregated embeddings, which will be integrated into the output representations of the current layer. To make training efficient, a two-stage training strategy is used where mention encoder is trained first and TOME is trained secondly. Evaluations are performed FEVER, HoVer, TriviaQA and ComplexWebQuestions, comparing with several baselines covering BERT, EaE, RAG and REALM, where TOME achieved consistent improvements on most of them.

**Summary Of The Review:**

The paper is clearly written and easy to follow. The motivation of the proposed method is intuitive. The model design is reasonable. The experiments are solid. In general, this is a good paper. One suggestion is to add more entity-insensitive task, to verify that the proposed model will not decrease its performances on such tasks.

---

> ### Author Response · Authors · 2021-11-16
> **Author Response**
>
> We thank the reviewer for their feedback and suggestions, and appreciate their positive comments.  We address the comments below.
>
> **NER**
>
> In general, we do not expect performance to be very sensitive to NER errors. We do not make use of entity types, only the presence of noun phrases and their approximate boundaries. We already use different NER during pre-training and fine-tuning - for pre-training we use the Google Cloud API and for fine-tuning we apply simple noun phrase chunking with spaCy. These methods exhibit very different behavior, but we compared Google Cloud vs spaCy on TriviaQA and performance was similar.
>
> **Entity-sensitive Tasks**
>
> We think that applying Mention Memory to a task that is unrelated to named entities is a very interesting suggestion. A natural candidate is commonsense reasoning; generating a memory of commonsense knowledge, and attending over this commonsense memory for a commonsense reasoning task. In a related case, the DrFact model (https://arxiv.org/abs/2010.14439) extends DrKIT (https://arxiv.org/abs/2002.10640) from a Wikipedia vKB to a commonsense vKB. We note that in that work the method from DrKIT generalized well to commonsense reasoning. We are certainly interested in performing such an extension, but view that as out of scope for this submission.

---

> > ### Author Response · Authors · 2021-11-19
> > **Follow-up**
> >
> > Hi reviewer giXh, we appreciate your time spent reviewing our manuscript. As the interactive discussion period is ending soon, we'd like to make sure we addressed your concerns as much as possible. In addition to our direct response to your concerns, our general response describes added experiments to strengthen empirical verification for TOME. Could you please let us know if you are satisfied with our response or if there are any remaining issues and/or if there are any additional clarifications we can provide?

---

### Official Review · Reviewer_ShxU · 2021-11-02

**Correctness:** 4
**Technical Novelty And Significance:** 3
**Empirical Novelty And Significance:** 3
**Recommendation:** 8
**Confidence:** 4

**Main Review:**

STRENGTHS

- The idea to attend over mentions rather than entities during feedforward is ambitious and refreshingly novel in the context of existing methods that focus on retrieving KB entries.

- The paper develops a staged training scheme to make training feasible, which would be otherwise infeasible even with the compute resource the authors have.

- TOME clearly outperforms EaE on the considered datasets with fewer parameters.


WEAKNESSES

- While TOME is certainly a neat extension of EaE, its main ideas are already pitched in previous works. Specifically, the benefits of virtual KBs (VKBs) over KBs, and performing multiple rounds of KB cross-attention within Transformer, are already highlighted in previous works. TOME's contribution is putting them together in a new model, but the main ideas individually are not a contribution of this work, so its novelty should be taken with a grain of salt.

- While I understand that TOME is small and fast, and larger explicit retrieval-based models have an advantage over TOME, it substantially lags behind the existing state-of-the-art models on both fact checking and QA. I don't think the performance shouldn't be a reason to dismiss the contributions of the work, but it does mean that TOME is not state-of-the-art.

- I think the paper can do better in justifying the choice of architecture and objectives. It's certainly reasonable, but there's no analysis. We don't know the impact of different span encoding architectures or pretraining objectives. Given how expensive it is to train TOME I suppose understanding development choices is a bit challenging.


**Summary Of The Paper:**

The paper proposes TOME, a Transformer that additionally performs cross-attention over contextual mention encodings (in 1 or 2 layers - TOME-1/TOME-2) along with the usual self-attention. It can be viewed as extending EaE (Fevry et al., 2020) from entities to mentions (aka., virtual KB - Dhingra et al., 2020). TOME is pretrained on 150m mentions extracted from Wikipedia by masked language modeling and other auxiliary objectives. It significantly outperforms EaE on fact checking (HoVer, FEVER) and entity-only QA (TriviaQA, CWQ), but lags behind explicit retrieve-and-read models like RAG, REALM, FiD.

**Summary Of The Review:**

The paper presents a large-scale pretrained Transformer that attends to mention encodings (EaE+VKB). It improves over EaE but lags behind state-of-the-art.

---

> ### Author Response · Authors · 2021-11-16
> **Author Response**
>
> We thank the reviewer for the comments and suggestions. We address them below.
>
> **Novelty**
>
> We agree that the reviewer has accurately characterized our proposed method - TOME can be viewed as extending EaE to use a vKB as introduced in DrKIT. As the reviewer has noted, past work has explored attention over entity information inside a Transformer and the use of vKBs, but not jointly. We believe that combining these ingredients is novel and highly nontrivial, and leads to qualitatively different behavior. Effectively scaling up model attention from Entities as Experts to a Wikipedia-sized corpus is also not straightforward, requiring in-model approximate nearest neighbor search and a novel pre-training scheme.
>
> **Results**
>
> We acknowledge that the performance of TOME lags behind RAG/FID. First, we would like to note that these models have more parameters and supervision. For example, the RAG retriever was pre-trained on gold TriviaQA evidence passages. However, we agree this is a reasonable criticism, especially since one of the advantages of our current approach is that it is cheaper to train a larger model. We intend to extend the method to much larger models in future work, but view that as out of scope for the current submission.
>
> In this work, we believe we show convincing evidence that TOME works better than a retrieve-and-read model with a same sized reader and similar supervision (REALM). At a high level, we observe a consistent pattern: TOME works about as well or slightly better than REALM for simple tasks with a single gold evidence passage, but achieves significantly higher performance for tasks which require evidence from multiple sources or otherwise difficult to retrieve evidence.
>
> TOME slightly outperforms REALM on FEVER. We compare TOME with REALM on the subset of TOME-answerable questions (see updated Table 2), and here TOME also slightly outperforms REALM. For HoVeR and ComplexWebQuestions which require evidence from multiple sources and/or hops, TOME strongly outperforms REALM.
>
> To further demonstrate this pattern, we evaluate on two additional datasets with challenging retrieval. The first is FoolMeTwice (FM2) from https://arxiv.org/abs/2104.04725. In FM2 claims are generated through an adversarial game, leading to more challenging retrieval. TOME also strongly outperforms REALM on this dataset (Table 1).
>
> The second is EntityQuestions, a new dataset introduced in https://arxiv.org/abs/2109.08535 which consists of questions involving rare entities. TOME outperforms REALM and dataset baselines by a large margin (see updated Table 2).
>
> We would have liked to compare with RAG on these more challenging datasets as well, but as we do not have access to the original RAG implementation this turned out to be nontrivial.
>
> **Analysis**
>
> We completely agree with the reviewer that it is important to form a better understanding of how choices in architecture and objective affect TOME’s behavior. As the reviewer notes, exhaustive analysis is difficult due to computational constraints, but we have added a number of ablations to appendix B.5 that help understand the impact of some of our design choices.
>
> First we investigate to what extent entity coreference loss is necessary. Table 9 in Appendix B.5 shows the effect of that the entity coreference loss is not important for TOME pre-training. This is unexpected, given that we approach TriviaQA and ComplexWebQuestions  as entity prediction tasks in very similar fashion to the pre-training objective. On the other hand, entity coreference loss is crucial for pre-training the BATCH-TOME mention encoder, consistent with our intuition that a semantically challenging tasks incentivizes the model to store useful information in memory.
>
> Next, Appendix B.6 also studies the effect of initialization. We find that, if TOME is initialized from BERT or from scratch rather than from BATCH-TOME, TOME does not learn to use the memory. In fact, TOME has to be initialized from the same BATCH-TOME used to generate the memory. This implies that multi-stage training is an important ingredient to make TOME succeed. Our explanation for why TOME is sensitive to initialization is that TOME needs to learn two skills: first, to effectively use retrieved mentions for its predictions, and second, to retrieve relevant mentions. Learning both capabilities end to end gives rise to a mutual dependence:  in order to get a signal for learning how to use retrieved mentions, the retrieved mentions have to be useful, and in order to learn to retrieve useful mentions the model needs to utilize retrieved mentions. If initialized from scratch the model cannot learn both capabilities at the same time. The pre-training stage with smaller in-batch memory functions as a curriculum.

---

> > ### Author Response · Authors · 2021-11-19
> > **Follow-up**
> >
> > Hi reviewer ShxU, we appreciate your time spent reviewing our manuscript. As the interactive discussion period is ending soon, we'd like to make sure we addressed your concerns as much as possible - could you please let us know if you are satisfied with our response or if there are any remaining issues and/or if there are any additional clarifications we can provide.

---

> > > ### Comment · Reviewer_ShxU · 2021-11-19
> > > **Thanks**
> > >
> > > Thanks a lot for the detailed response. I find the paper definitely stronger thanks to your additional results and ablation studies, and I've increased my score to accept.
> > >
> > > I personally support the idea of attending to KBs inside the model. But I think the truth is that right now (1) it's not as critical as we'd like to believe (at least in many settings), and (2) it's really difficult to train. I look forward to this idea getting refined in the future.
> > >
> > > One more comment: it'd be good to include the baseline results for FM2 and EQ in the original papers as reference points, since you're only comparing with your own run of EaE and REALM. It does seem that your top-1 accuracy of 66.0 in EQ is state-of-the-art (I assume this is with finetuning), but the original paper gives top-20 accuracies where BM25 gets 71.2, so it's a bit difficult to contextualize your results.

---

> > > > ### Author Response · Authors · 2021-11-19
> > > > **Follow-up**
> > > >
> > > > Thank you for your response!
> > > >
> > > > We agree that these models can be tricky to train, and hope to develop better principled training techniques for large-scale end-to-end retrieval.
> > > >
> > > > Including original FM2/EQ results makes sense. We included top-1 results in the main text to be more comparable with our other QA results, but we discuss top-20 results in appendix B.4/Table 7, showing strong performance improvements there also. We will add original FM2 baseline results to the appendix as well.

---

> > > > > ### Comment · Reviewer_ShxU · 2021-11-19
> > > > > **Oops**
> > > > >
> > > > > Ah I see, I was only looking at Table 8 and missing Table 7 for EQ. Thanks for the clarification! This is indeed a significant improvement.

---

### Official Review · Reviewer_nJu8 · 2021-11-02

**Correctness:** 3
**Technical Novelty And Significance:** 3
**Empirical Novelty And Significance:** 3
**Recommendation:** 8
**Confidence:** 4

**Main Review:**

### Pros
- This work extends the past work on knowledge-augmented LMs and proposes an approach to encode knowledge sources in a more fine-grained way.
- This architecture maintains explicit memories for mention spans in the knowledge source, so it still retains some interpretability.
- Strong results on HoVer and CWQ.
- The method and the training details are clearly explained.
- The authors will release the code and model checkpoints.

### Cons
- The model performance is inconsistent across different datasets. The TOME model shows strong results on HoVer but underperforms RAG on FEVER. Many FEVER claims are simple definitive sentences (e.g., single hop relations) and can be found in English Wikipedia, so it’s understandable that end-to-end retriever-based models might perform well. I wanted to see more analysis on this point (why is the number lower?). Similarly, it did well on CWQ, but the TriviaQA numbers are behind the retriever-based models. I think instance-level error analysis would be helpful to articulate underlying issues (from data or a model).
- Although the authors promise to release their code and checkpoints, I’m wondering if people with limited resources (e.g., researchers in academia) can use this model. In Appendix B, it says that the TOME model is finetuned on 32 TPU. From the current version of the paper, it’s unclear if this model can be fine-tuned on a fewer number of TPUs (or GPUs?).

### Questions / Suggestions
- It seems that the pretraining data is annotated using off-the-shelf tools (NER, EL). I’m wondering about the trade-off between the entity coverage (e.g., adding more mention spans on top of the hyperlinks) and the data quality (cascading errors from the NER, EL tools).
- If  the TOME model is only able to train on and answer 84% of examples, it might be interesting to see the baseline performance using the same train/eval sets.
- There might be other choice for claim verification such as FM2 (https://arxiv.org/pdf/2104.04725.pdf) , which consists of higher quality claims compared to FEVER (but the data size is smaller).
- The Fact as Experts paper is also related: https://arxiv.org/pdf/2007.00849.pdf

### Minor
- Use “English Wikipedia” in the main text (Bender rule).
- Passage length is denoted as $T$ in Section 2 Notation, but $L$ is used in Appendix A.


**Summary Of The Paper:**

__Method:__ This paper proposes a new approach to integrate knowledge sources into Transformer-based models. Concretely, entity mentions found in English Wikipedia (approx. 150M -- these mentions are linked to Wikipedia entities) are encoded into __Mention Memory__, which consists of key vectors and value vectors.  The knowledge representations in a __Mention Memory__ are accessed from a __TOME__ block, which is a stack of transformer blocks with a __memory attention__ layer. In this layer, a mention in the input sequence is converted into a “knowledge-injected” vector representation which is given by a weighted sum of the mention vectors (i.e., value vectors). The entire model (TOME) is pretrained in two stages. First, the Mention Encoder is pretrained, and the Mention Memory is generated. Here, the Mention Encoder is trained with a small-scale TOME architecture (Batch-TOME) using the masked LM loss and the coreference loss. Next, the Mention Memory is fixed, and other parameters in the TOME model are updated. The TOME model is trained with a masked LM task as well as an entity prediction task. This output layer allows us to use the TOME model for downstream tasks such as TriviaQA and Complex WebQA.

__Evaluation:__ This approach is evaluated on two tasks: claim verification (HoVer, FEVER) and QA (TriviaQA, CWQ) and compared with various baselines (EaE, RAG, REALM etc.). In the claim verification experiments, the TOME models (both 1 and 2 blocks) outperform all baselines on HoVer, which require reasoning using multiple sources. On FEVER, the TOME models outperform baselines except RAG. In the QA experiments, the TOME model with 2 blocks outperforms all baselines on CWQ. On TriviaQA, the TOME models outperform similar models such as EaE and much larger generative LMs (T5) but underperform the retriever-based models. Additionally, the authors perform analysis on retrieved passages, memory sizes, and performance on unseen entities in a QA task.


**Summary Of The Review:**

I am leaning towards acceptance. The way TOME infuses textual knowledge into Transformer models is new. This approach actually integrates a passage retriever into a transformer-based LM and potentially covers broad downstream tasks. But, I think the analysis on the experimental results can be improved.

---

> ### Author Response · Authors · 2021-11-16
> **Author Response**
>
> We thank the reviewer for carefully reviewing our work and providing extensive suggestions. We address the comments below.
>
> **Empirical comparison with retrieve-and-read**
>
> At a high level, we observe a consistent pattern: TOME works about as well or slightly better than REALM for simple tasks with a single gold evidence passage, but achieves significantly higher performance for tasks which require evidence from multiple sources or otherwise difficult to retrieve evidence.
>
> TOME slightly outperforms REALM on FEVER. We follow the reviewer’s suggestion and compare TOME with REALM on the subset of TOME-answerable questions (see the updated Table 2), and here TOME also slightly outperforms REALM. For HoVeR and ComplexWebQuestions which require evidence from multiple sources and/or hops, TOME strongly outperforms REALM.
>
> To further demonstrate this pattern, we evaluate on two additional datasets with challenging retrieval. The first is FoolMeTwice (FM2), a fact verification dataset suggested by the reviewer. In FM2 claims are generated through an adversarial game, leading to more challenging retrieval. TOME also strongly outperforms REALM on this dataset (see the updated version of Table 1).
>
> The second is EntityQuestions, a new dataset introduced in https://arxiv.org/abs/2109.08535 which consists of questions involving rare entities. TOME outperforms REALM as well as the dataset baselines on this dataset by a large margin (see Table 2).
>
> We would have liked to compare with RAG on these more challenging datasets as well, but as we do not have access to the original RAG implementation this turned out to be nontrivial. We would like to note that RAG has two major advantages over TOME: first, it uses a BART-Large decoder, which is four times the size of our reader model, and second, its retrieval module is pre-trained on gold retrieval for QA data, including gold retrievals for TriviaQA. Increasing the size of the reader and using supervised pre-training data are reasonable choices to improve performance that can also be applied to the TOME architecture, but we view that as out of scope for this work.
>
> Especially on binary claim verification datasets, for which there is no difference between answering abstractively or with a classification layer, REALM is essentially equivalent to RAG with a BASE-sized reader and without supervised pre-training. This makes REALM a fairer choice as a retrieve-and-read comparison for TOME.
>
> **NER/Entity Linking**
>
> In general, we do not expect performance to be very sensitive to EL/NER methods. For one, we already use different NER during pre-training and fine-tuning - for pre-training we use the Google Cloud API and for fine-tuning we apply simple noun phrase chunking with spaCy. These methods exhibit very different behavior, but we compared Google Cloud vs spaCy on TriviaQA and performance was similar.
>
> We added an ablation of the entity linking objective to Appendix B.5 and Table 9. Note that the entity linking objective is not important for TOME pre-training, only for Batch-TOME pre-training. Moreover, during pre-training we only use a subset of the memory, so reduced recall of the entity linker would not affect memory size during pre-training.
>
> **Fine-tuning requirements**
> The model can be fine-tuned with full memory on a server of 8 A100 GPUs or 16 v3/v4 TPUs. A model with half memory (75M mentions) can be fine-tuned on 8 V100s/P100s or 8 TPUs. We think this is sufficiently viable to be interesting for academic labs, though we are aware not all labs will have access to such resources. We have added this information to the appendix.
>
> **Facts as Experts**
> Facts as Experts is indeed highly related! It was eventually published under the name FILM (Fact Injected Language Model), at https://aclanthology.org/2021.naacl-main.288/. Section 3 describes the main differences between TOME and FILM: TOME operates over text rather than a symbolic knowledge base, retrieves fine-grained mention representations as values rather than aggregate entity representations, and reasons over the retrieved information within the Transformer.
>
> **Additional**
> Thank you for pointing out the Bender violation! We have added English Wikipedia to the main text.

---

> > ### Comment · Reviewer_nJu8 · 2021-11-17
> > **Thanks for the comprehensive response**
> >
> > I would like to thank the authors for the additional analyses and their comprehensive response to my comments. I agree that REALM is the reasonable competitor of TOME. TOME consistently outperforms REALM on three different types of claim verification datasets. On QA datasets, TOME is comparable with REALM on the standard TQA but outperforms REALM on TQA examples with Wikipedia entities (84% subset). TOME also outperforms REALM on the other QA datasets that require complex reasoning (CWQ) or challenging retrieval involving rare entities (EQ). These results clarify my questions about the empirical evidence, and the authors have updated their paper to address those points. Given these, I revised my rating to reflect that.
> >
> > Minor:
> > - Section 4.4 (1st paragraph)  *... In order to compare head-to-head performance performance, ...*

---

### Official Review · Reviewer_djR9 · 2021-11-04

**Correctness:** 4
**Technical Novelty And Significance:** 3
**Empirical Novelty And Significance:** 4
**Recommendation:** 8
**Confidence:** 4

**Main Review:**

Strengths:
* While it's true that this approach combines aspects of previous work and is therefore not extremely novel, the combination is interesting and well motivated, and the paper is technically sound. This approach reminds me a lot of REALM and RAG, with the crucial difference that the retrieved knowledge isn't concatenated to the inputs but rather directly comes in the form of dense representations. This has a series of advantages, which the authors explain well in section 3 – no length constraint, less computationally expensive.
* The paper is easy to follow. The appendices provide useful information on the pretraining procedure.
* Performance on four standard tasks is convincing and the choice of baselines is reasonable.

Weaknesses:
* While pretraining is described very clearly, with full details of hyperparameters given in an appendix, details on the finetuning procedure are very light.
* I would have liked to see a part of section 3 dedicated to discussing generative models like RAG.

Typos:
* p10: "we have release"

**Summary Of The Paper:**

The authors propose to extend the standard Transformer architecture by allowing it to attend over factual information, represented as a large memory of dense representations of entity mentions. The resulting architecture is made up of two parts: a mention encoder, which is used to build up the "mention memory"; and the transformer model augmented with attention over the memories. These are pre-trained in two stages for efficiency reasons. The model is evaluated on two claim verification datasets and two QA datasets, showing convincing performance against comparable methods.

**Summary Of The Review:**

A strong, well written paper describing a novel approach for incorporating knowledge into a transformer encoder. The main components are not highly novel from a technical point of view, but their combination is.

---

> ### Author Response · Authors · 2021-11-16
> **Author Response**
>
> We thank the reviewer for taking the time to carefully review our work, and appreciate the positive comments and suggestions. We have expanded our description of the fine-tuning procedure in Appendix B.1-B.4, and added a discussion of generative models such as RAG and FiD to Section 3.

---

> > ### Comment · Reviewer_djR9 · 2021-11-21
> > **Thanks**
> >
> > Thank you for addressing my comments, especially for adding more details about fine-tuning. I am satisfied with your responses, and I believe you have addressed the comments by reviewr giXh in a satisfactory way. As a result, I confirm my initial assessment and I would be happy to see this paper accepted.

---

### Author Response · Authors · 2021-11-16
**General response to reviewer comments**

In response to reviewer comments and suggestions, we have made the following high-level changes to the paper.

First, we added results on two challenging new datasets: the adversarially generated FoolMeTwice claim verification dataset (https://arxiv.org/abs/2104.04725) and the EntityQuestions dataset based on rare entities (https://arxiv.org/abs/2109.08535). The results can be found in Table 1 and 2. In both cases, TOME strongly outperforms REALM and Entities as Experts, and in the case of EntityQuestions, sets the state of the art (see Table 7).

Second, we added a TriviaQA setting where we train and evaluate only on questions for which answers are Wikipedia entities (and are therefore answerable by TOME and Entities as Experts). TOME slightly outperforms REALM in this setting. Our results are consistent with the intuition that TOME and REALM perform similarly on tasks with straightforward retrieval, but TOME strongly outperforms REALM on tasks with challenging retrieval.

Finally, we added ablations (Appendix B.5 and B.6) studying the effect of pre-training objectives and TOME initialization on performance.

---

### Decision · Program_Chairs · 2022-01-20

**Decision:**

Accept (Poster)

**Comment:**

The paper proposes TOME, which extends Transformer by attending to entity mention memory. Experiments are conducted on claim verification and QA.

Reviewers generally found the paper is solid. However, the novelty appears to be limited and is mainly in the combination of existing models.